# Doxycycline and Minocycline Act as Positive Allosteric Modulators of the PAC1 Receptor and Induce Plasminogen Activators in RT4 Schwann Cells

**Sarah Thomas Broome** [1] , **Giuseppe Musumeci** [2] **and Alessandro Castorina** [1,*]

1   Laboratory of Cellular and Molecular Neuroscience (LCMN), School of Life Sciences, Faculty of Science, University of Technology Sydney, Sydney, NSW 2007, Australia; sarah.j.thomasbroome@student.uts.edu.au
2   Department of Biomedical and Biotechnological Sciences, Anatomy, Histology and Movement Sciences Section, School of Medicine, University of Catania, 95125 Catania, Italy; g.musumeci@unict.it
*   Correspondence: Alessandro.Castorina@uts.edu.au

**Featured Application: Targeting the PAC1-receptor has the potential to exert neuroprotective effects against a range of neurotoxic, neurodegenerative and inflammatory insults. The novel discovery that doxycycline and minocycline act as positive allosteric modulators of the PAC1 receptor provides an exciting opportunity to therapeutically target this receptor. This study provides evidence for the ability of these tetracyclines to promote Schwann cells activities associated with improved axonal regeneration through the induction of a PAC1-mediated proteolytic activity of tPA and uPA.**

**Abstract:** Regeneration of peripheral nerves depends on the ability of axons to navigate through an altered extracellular environment. It has been suggested that Schwann cells facilitate this process through their secretion of neuropeptides and proteases. Using the RT4-D6P2T Schwann cell line (RT4), we have previously shown that RT4 cultures endogenously express the neuropeptide PACAP, and respond to exogenous stimulation by inducing the expression of tissue plasminogen activator (tPA) and urokinase plasminogen activator (uPA) via PAC1 receptor activation. In this study, based on recent findings showing that doxycycline and minocycline act as positive allosteric modulators (PAMs) of the PAC1 receptor, we tested if treatment with these tetracyclines induced the expression and activity of tPA and uPA in RT4 cells. Using ELISA and zymographic analyses, we demonstrate that doxycycline and minocycline reliably induce the secretion and activity of both tPA and uPA, which is paralleled by increased expression levels, as shown by immunocytochemistry and Western blots. These actions were mediated, at least in part, by the PAC1 receptor, as PACAP6-38 mitigated tetracycline-induced expression and activity of tPA and uPA. We conclude that doxycycline and minocycline act as PAMs of the PAC1 receptor to promote proteolytic activity in RT4 cells.

**Keywords:** pituitary adenylate cyclase-activating peptide (PACAP); PAC1 receptor; doxycycline; minocycline; positive allosteric modulator; Schwann cells; tissue plasminogen activator (tPA); urokinase plasminogen activator (uPA)

## 1. Introduction

Peripheral nerve injury can arise from systemic diseases or localized damage; however, unlike the central nervous system (CNS), the peripheral nervous system (PNS) displays a high regenerative capacity [1]. This is mainly driven by Schwann cells, the myelinating glia of the PNS, which display high plasticity following peripheral nerve injury. Upon nerve injury, Schwann cells are spontaneously reprogrammed to switch from a myelinating phenotype to a progenitor-like state that is able to promote nerve regeneration [2]. Part of this process includes the secretion of proteases such as plasminogen activators, whose proteolytic activities clear any residual cell debris and altered extracellular matrix, creating

a clear nerve conduit to allow axon regeneration [3,4]. The two major plasminogen activators involved in this process are the tissue plasminogen activator (tPA) and urokinase plasminogen activator (uPA) [3].

There is evidence that the neuropeptide pituitary adenylate cyclase-activating peptide (PACAP) is also involved in peripheral nerve regeneration [5,6]. PACAP is widely distributed throughout the central and peripheral nervous systems and is associated with a myriad of biological functions including antiapoptotic, neuroprotective and anti-inflammatory [7–10]. The activity of PACAP is mediated by three G protein-coupled receptors (GPCRs) PAC1, VPAC1 and VPAC2. Whereas PACAP binds with a high affinity to these three receptors, it shows a much higher affinity (>100-fold) for the PAC1 receptor [11]. As such, the PACAP preferring receptor PAC1 is thought to mediate most of the antiapoptotic and neuroprotective effects of PACAP. Based on these findings, the PACAP/PAC1 axis has been investigated as a potential target for the treatment of neurological diseases, including Alzheimer's disease [12], stroke [13], headaches (using PAC1 antagonists) [14] and other neurological disorders (reviewed in [15–18]).

Emerging studies suggest that the PAC1 receptor might promote the regeneration and functional recovery following peripheral nerve injury, as some reports have shown that both PACAP and PAC1 are upregulated after nerve injury [19]. Moreover, it has been demonstrated that axon regeneration is strongly reduced in PACAP knockout animals [20]. In vitro, we have previously demonstrated that RT4 Schwann cell-like cultures express both PACAP and PAC1, and stimulation of these cells with PACAP promoted tPA and uPA expression and activity [21,22]. Additionally, we demonstrated that BDNF mimicked PACAP-induced up-regulation of tPA expression and activity in these cell lines [22].

Recently, it has been shown that doxycycline and minocycline act as positive allosteric modulators (PAMs) of the PAC1 receptor [23]. Doxycycline and minocycline are tetracyclines, broad spectrum antibiotics that share similar structures and functions [24]. Both tetracyclines have shown promising potential as therapeutic targets for neurological diseases due to their high lipophilic nature that allows them to readily cross the blood–brain barrier and other biological membranes [25]. Both doxycycline and minocycline have been shown to exert neuroprotection and anti-inflammatory activities in a range of neurological pathologies, similarly to PACAP [25–29]. Most importantly, it has been shown that the neuroprotective activity of doxycycline is enhanced by the activation of the PAC1 receptor [30].

In the present study, we investigated the novel role of doxycycline and minocycline as potential PAMs of PAC1 receptor in Schwann cell lines. To achieve this goal, we tested if exogenous administration of either tetracycline would induce tPA and/or uPA expression and activity in a similar way as exogenous PACAP. In addition, we assessed if PAC1 receptor blockade prevented these effects in RT4 Schwann cell-like cultures.

## 2. Materials and Methods

### 2.1. RT4 Schwann Cell Line

This study used the rat Schwann cell line RT4-D6P2T (ATCC number CRL-2768) obtained from the American Type Culture Collection (Rockville, MD, USA). Cells were cultured in Dulbecco's Modified Eagle's Medium (DMEM) supplemented with 10% heat-inactivated fetal bovine serum (FBS) and 100 U/mL penicillin, and 100 µg/mL streptomycin (Sigma-Aldrich, Castle Hill, NSW, Australia). Cells were incubated at 37 °C in a humidified environment with 5% $CO_2$. Cells were grown to 80–85% confluence and treated as indicated in the corresponding experiments.

### 2.2. ELISA tPA and uPA Activity

The concentrations of uPA and tPA were determined by Enzyme Linked Immune Sorbent Assay (ELISA) commercial kits (Abcam, Cambridge, MA, USA; tPA [ab198510] and uPA [ab108917]) according to the manufacturer's instructions. RT4 cells were either left untreated (controls) or treated with increasing concentrations of either doxycycline or

minocycline (10, 50, 100, 500 and 1000 ng/mL) and tPA and uPA levels assessed 24 h later. Each condition was tested using five different batches of cells (n = 5/group).

### 2.3. Cell Viability Assay

To assess cell viability, we used the cell proliferation kit I (Sigma-Aldrich, NSW, Australia). Cells were treated with increasing doses of either doxycycline or minocycline for 24 h as above. 10 μL of MTT labeling reagent was added to each well and incubated for 4 h. 100 μL of solubilization solution was added to each well and incubated overnight at 37 °C. Absorbance was measured at 565 nm in the TECAN infinite M1000-PRO ELISA reader (ThermoFisher Scientific, Victoria, Australia).

### 2.4. Zymography of tPA and uPA Activity

The activity of tPA and uPA was determined by zymographic analysis using a procedure previously published [22,31]. Briefly, following indicated treatment, cells were collected with 1% Triton X-100 in PBS buffer. Equal amounts of protein were loaded onto a 12% polyacrylamide gel containing 2 mg/mL casein in the presence of 5 μg/mL plasminogen. After electrophoresis, enzyme reaction was initiated by incubating the gel in 0.1 M glycine-NaOH (pH 8.3) at 37 °C for 18 h, and lytic areas were identified after staining the gel with a solution of 30% methanol, 10% glacial acetic acid and 0.5% Coomassie blue G250. The gels were then de-stained and scanned with Bio-Rad Imaging system. Images were assessed semi-quantitatively using ImageJ software. To exclude interference by matrix metalloproteinase, EDTA (2 mM) was included in the glycine-NaOH buffer during incubation. Gels that did not contain plasminogen were also run and failed to produce lytic areas corresponding to plasminogen activators (data not shown). Bands were identified based on their relative molecular weight.

### 2.5. Immunocytochemistry

To determine the expression of tPA in RT4 cells, immunocytochemistry was performed. Briefly, $1 \times 10^4$ cells were seeded in poly-L-lysine (Sigma-Aldrich, Castle Hill, NSW, Australia) pre-coated tissue culture coverslips (22 mm Ø, Sarstedt, SA, Australia). Cells were fixed with 4% filtered paraformaldehyde (PFA: 4% in PBS pH 7.4) (Sigma-Aldrich, Castle Hill, NSW, Australia). Coverslips were then washed three times with ice cold PBS. Cell were then permeabilized for 10 min in PBS containing 0.25% Triton X-100 (Sigma-Aldrich, Castle Hill, NSW, Australia), followed by 3 × washes in PBS for 5 min. Thereafter, cells were soaked in 1.5% $H_2O_2$/PBS solution for 15 min to quench endogenous peroxidases. Non-specific binding of antibodies was then prevented by incubating coverslips with 1% BSA in PBST for 30 min. Once completed, the cells were incubated in diluted rabbit anti-tPA primary antibody (Abcam, Cambridge, MA, USA [ab157469]; diluted 1:500 in PBST and 1% BSA) in a humidified chamber overnight at 4 °C. The next day, the primary antibody was removed with 3 × washes in PBS for 5 min. Cells were then incubated with secondary antibodies (goat antirabbit IgG H&L (HRP) [ab97051] at 1/5000 dilution) in 1% BSA in PBST for 2 h at room temperature with gentle oscillation. Secondary antibody solution was then removed, and cells were washed again three times with PBS for 5 min. A drop streptavidin peroxidase was then added to each well (containing the coverslip) and allowed to incubate for 30 min at room temperature, followed by 3 × 5 min washes in PBST on an orbital shaker. 3,3′-Diaminobenzidine (DAB) substrate (Sigma-Aldrich, Castle Hill, NSW, Australia) was added to each cell coverslip under a fume hood. Once the cells started turning brown, the reaction was stopped with 2 × washes in PBS for 5 min each on a shaker. Counterstaining of nuclei was performed by dipping each coverslip into a staining dish of hematoxylin for 10 s, followed by an acid bath (200 mL UltraPure water and 1–3 drops of acetic acid). Stained coverslips were then imaged using a Nikon Eclipse TS2 inverted microscope (magnification 20×). Scale bar is 60 μm.

### 2.6. Western Blot

Protein lysate were homogenized in RIPA buffer (Sigma-Aldrich, St. Louis, MO, USA), which was supplemented with cOmplete™ULTRA protease inhibitor cocktail (Roche Life Science, North Ryde, NSW, Australia). Lysates were then sonicated twice at 50% power for 10 s using an ultrasonic probe, followed by centrifugation prior to protein quantification. Protein concentrations were determined using the Bicinchoninic Acid (BCA) Assay Kit (ThermoFisher Scientific, Waltham, MA, USA).

Lysates (20 μg) were separated by sodium dodecyl sulfate (SDS)-polyacrylamide gel electrophoresis (SDS-PAGE) using 4–20% Mini-PROTEAN® TGX Stain-Free™ protein gels (15 wells) (Bio-Rad, Hercules, CA, USA). Samples were prepared by 3.75 μL of 4× Laemmli buffer (Bio-Rad, Hercules, CA, USA) and β-mercaptoethanol (Sigma-Aldrich, St. Louis, MO, USA) mix (Laemmli buffer 20:1 with β-mercaptoethanol) to a final volume of 15 μL. Samples were then denatured at 70 °C for 10 min using the T100™ thermal cycler.

Samples were resolved on gels and transferred to a PVDF membrane using the semi-dry approach (Trans-Blot Turbo™ Transfer Pack; Bio-Rad, Hercules, CA, USA). Once proteins were transferred, the membrane was removed from the apparatus and briefly washed in tris-buffered saline containing Tween®20 (TBST) 1 × three times before blocking with 5% blocking buffer containing 5% skim milk in TBST 1 × for 1 h at room temperature at 55–60 rpm. The membranes were probed with antirabbit primary antibodies recognizing either tPA (1:1000; GeneTex, Irvine, CA, USA; cat. No. GTX103453), uPA (1:750; LifeSpan Biosciences Seattle, WA, USA; cat No. LS-C193095) or GAPDH (1:1000; BioRad, Hercules, CA, USA; cat. No. VPA00187) (used as loading control) overnight at 4 °C under gentle oscillation (55 rpm). Following incubation, membranes were washed thoroughly. Finally, membranes were incubated with a goat antirabbit secondary antibody (1:10,000; BioRad, Hercules, CA, USA; cat. No. STAR208P) for 1h at room temperature and washed again before imaging. To visualize immunoreactive bands, we used Clarity™ Western ECL Blotting Substrate (Bio-Rad, Hercules, CA, USA). Images were acquired on the ChemiDoc™ MP System (Bio-Rad, Hercules, CA, USA). Band intensities were quantified using ImageJ software and values were normalized to GAPDH.

### 2.7. Statistical Analysis

Statistical analysis was performed using GraphPad Prism version 7.04 for Windows, GraphPad Software, San Diego California, USA, www.graphpad.com (accessed on the 12 January 2021). All experimental data are reported as mean ± S.E.M. To assess statistical differences between three or more groups we used one-way analysis of variance (ANOVA) followed by Tukey's *post-hoc* test, unless otherwise stated. $p$-values $\leq 0.05$ were considered statistically significant.

## 3. Results

### 3.1. Dose–Response Effects of Doxycycline and Minocycline on tPA and uPA Secretion in RT4 Cells

To investigate if doxycycline and minocycline were able to stimulate the secretion of tPA and uPA by RT4 Schwann cell-like cultures, we measured the amount of plasminogen activators released in culture media by enzyme-like immunosorbent assay (ELISA) after treatment with either antibiotic for 24 h. We utilized increasing concentrations of doxycycline and minocycline (0, 10, 50, 100, 500 and 1000 ng/mL) to establish the optimal concentration to use in subsequent experiments (Figure 1). PACAP, which was used as a positive control, significantly increased tPA (*** $p < 0.001$ vs. Control) and uPA (* $p < 0.05$ and ** $p < 0.01$) levels in cultured media, confirming the dose- and time-dependent increase in plasminogen activators expression and activity of PACAP in RT4 cells in earlier studies [22].

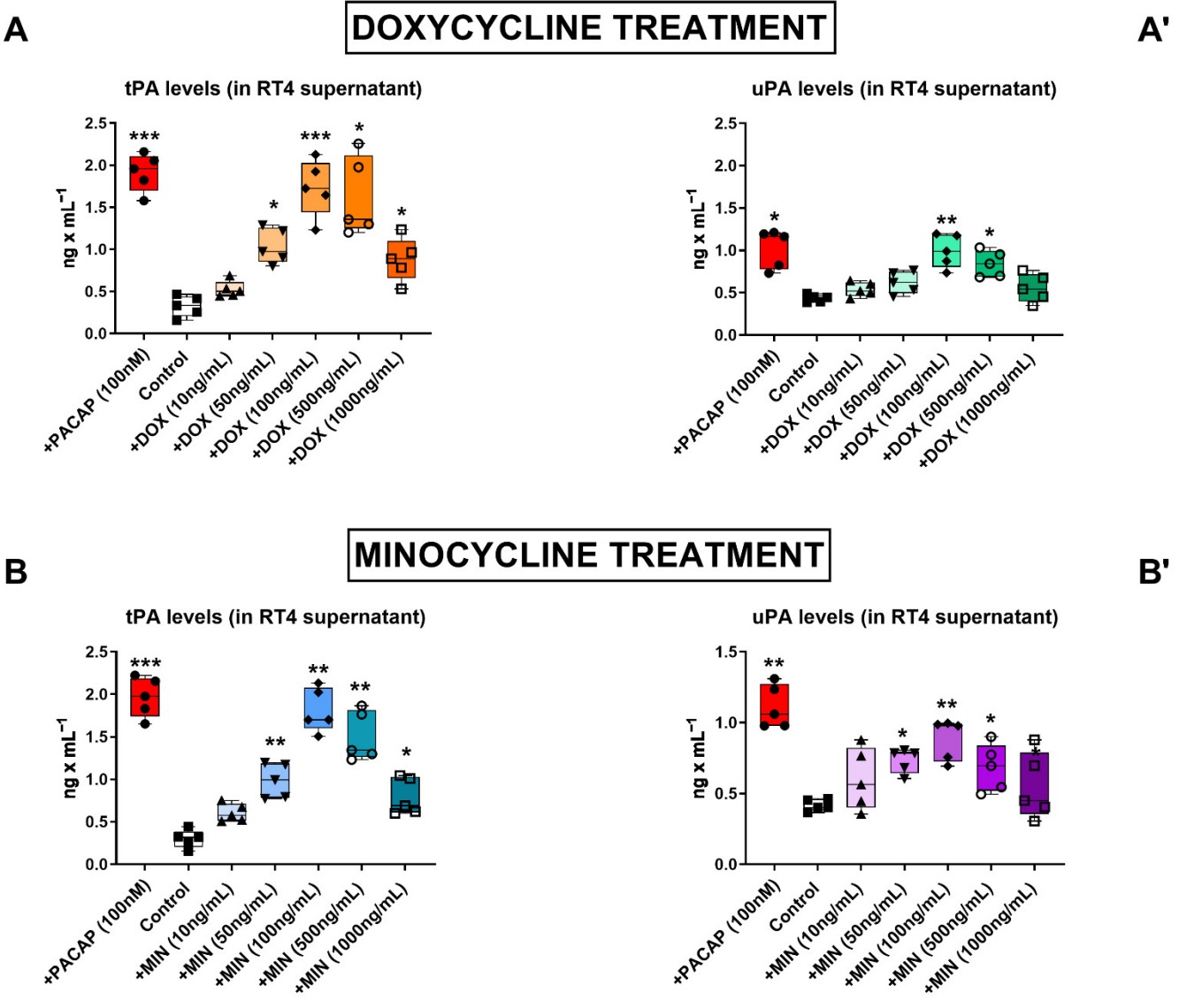

**Figure 1.** Dose–response effect of doxycycline and minocycline on tPA and uPA secretion in RT4 cells. (**A**,**B**) Levels of secreted tPA or uPA in the supernatant of RT4 cells treated with either doxycycline (**A**,**A′**) or minocycline (**B**,**B′**). RT4 cells were treated with the indicated treatments for 24 h. Cell supernatants were harvested and the concentration of tPA and uPA were determined by ELISA. Data shown are the result of two independent experiments using five different batches of cells (n = 5). * $p < 0.05$, ** $p < 0.01$ or *** $p < 0.001$, as determined by ANOVA followed by Dunnett's *post-hoc* test.

Experimental data demonstrated that both doxycycline and minocycline dose-dependently increased the levels of secreted tPA and uPA in the supernatant of RT4 cells, although uPA induction was less remarkable than tPA (Figure 1A,B,A′,B′, respectively). Doxycycline reliably increased tPA secretion at 50 ng/mL (* $p < 0.05$ vs. Control, Figure 1A), however, 100 ng/mL doxycycline was required to significantly increase uPA levels as well (** $p < 0.01$ vs. Control, Figure 1A′).

Minocycline treatment significantly increased tPA (** $p < 0.01$ vs. Control, Figure 1B) and uPA levels (* $p < 0.05$ vs. Control, Figure 1B′) in the culture media at 50 ng/mL; however, similarly to doxycycline, minocycline activity also peaked at 100 ng/mL (** $p < 0.01$ vs. Control for both tPA and uPA). Accordingly, we identified 100 ng/mL as the most effective concentration of antibiotic to use in subsequent experiments.

### 3.2. Effects of Doxycycline or Minocycline Treatment on RT4 Cell Viability

To investigate if doxycycline or minocycline were toxic to cells, we treated RT4 cells with increasing concentrations of either doxycycline or minocycline and assessed viability using the MTT assay. Doxycycline (DOX) and minocycline (MIN) were not toxic to RT4 cells up to the concentration of 100 ng/mL, where cells displayed normal growth rate (**** $p < 0.0001$ vs. Ctrl for both DOX and MIN). Treatment with higher concentrations of either antibiotic (500 and 1000 ng/mL) caused a progressive decline in cell viability, which significantly declined at 1000 ng/mL (# $p < 0.05$ vs. 100 ng/mL DOX or MIN, respectively, Figure 2A,B).

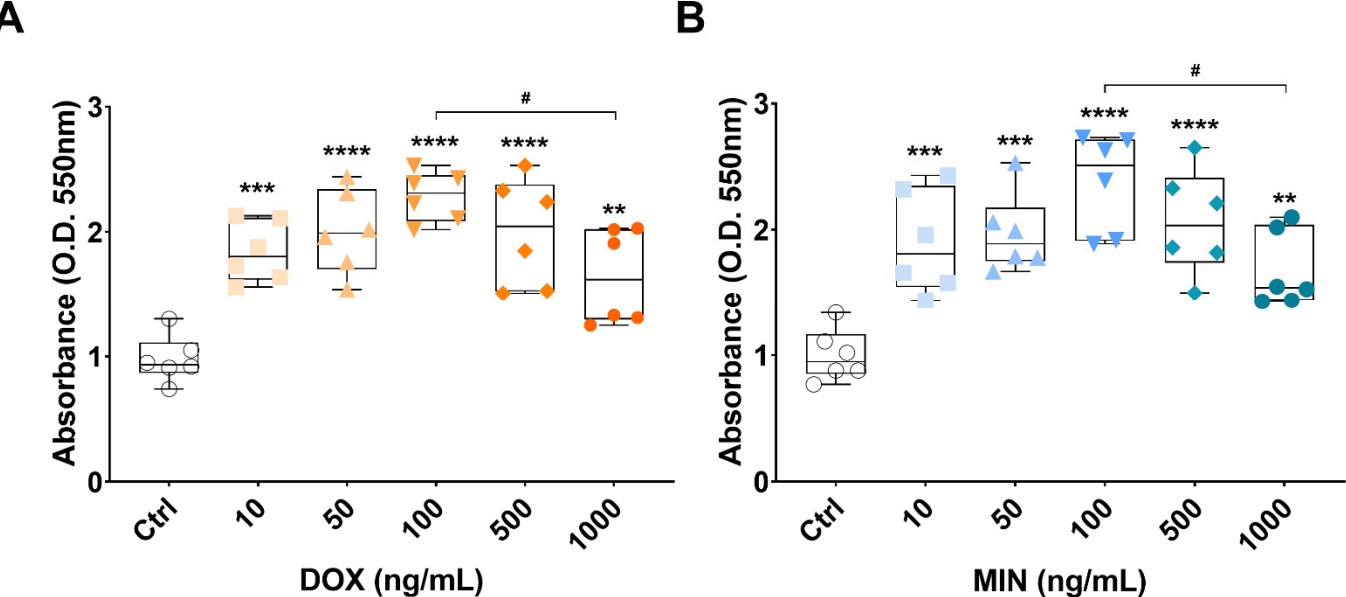

**Figure 2.** Cell viability of RT4 cells exposed to increasing concentrations of doxycycline or minocycline. Analysis of cell viability (MTT assay). RT4 cells were grown under normal conditions and exposed to increasing concentrations (0, 50, 100, 500 or 1000 ng/mL) of (**A**) doxycycline or (**B**) minocycline for 24 h. Values are reported as mean optical densities (OD) ± S.E.M. from two experiments, each using five batches of cells. ** $p < 0.01$, *** $p < 0.001$ or **** $p < 0.0001$ vs. Ctrl; # $p < 0.05$ vs. 100 ng/mL DOX (**A**) or MIN (**B**), as determined by ANOVA followed by Tukey's *post-hoc* test.

### 3.3. Enzymatic Activity of tPA and uPA in RT4 Cells Exposed to Doxycycline or Minocycline

Our next step was to establish if doxycycline or minocycline stimulated tPA and uPA enzymatic activity in RT4 cells using zymography. We used exogenous PACAP as our positive control [22]. RT4 cells were treated with either PACAP (100 nM), doxycycline or minocycline (100 ng/mL) for 24 h.

Quantification of both tPA and uPA lytic bands on gels (white bands corresponding to 70 kDa [tPA] and 50 kDa [uPA], Figure 3A) confirmed that exogenous PACAP increases the activity of both plasminogen activators (** $p < 0.01$ vs. Control). Both doxycycline and minocycline treatments significantly increased tPA (** $p < 0.01$ and *** $p < 0.001$ vs. Control, respectively, Figure 3B,B') and uPA activity (* $p < 0.05$ vs. Control for both, Figure 3B,B').

### 3.4. tPA-like Immunoreactivity in RT4 Cells Exposed to PACAP, Doxycycline or Minocycline

We have previously demonstrated that treatment with PACAP or the PAC1 agonist maxadilan did not affect the distribution but only the expression of tPA in RT4 cells [22]. As such, we performed immunocytochemistry to determine if, similarly to PACAP, doxycycline or minocycline treatment increased the expression but not the cellular distribution of either plasminogen activators. Sections where the primary antibody was omitted were used as negative controls (Figure 4A). Cells were either left untreated (Figure 4B) or exposed to PACAP (Figure 4C), doxycycline (Figure 4D) or minocycline (Figure 4E) for 24 h.

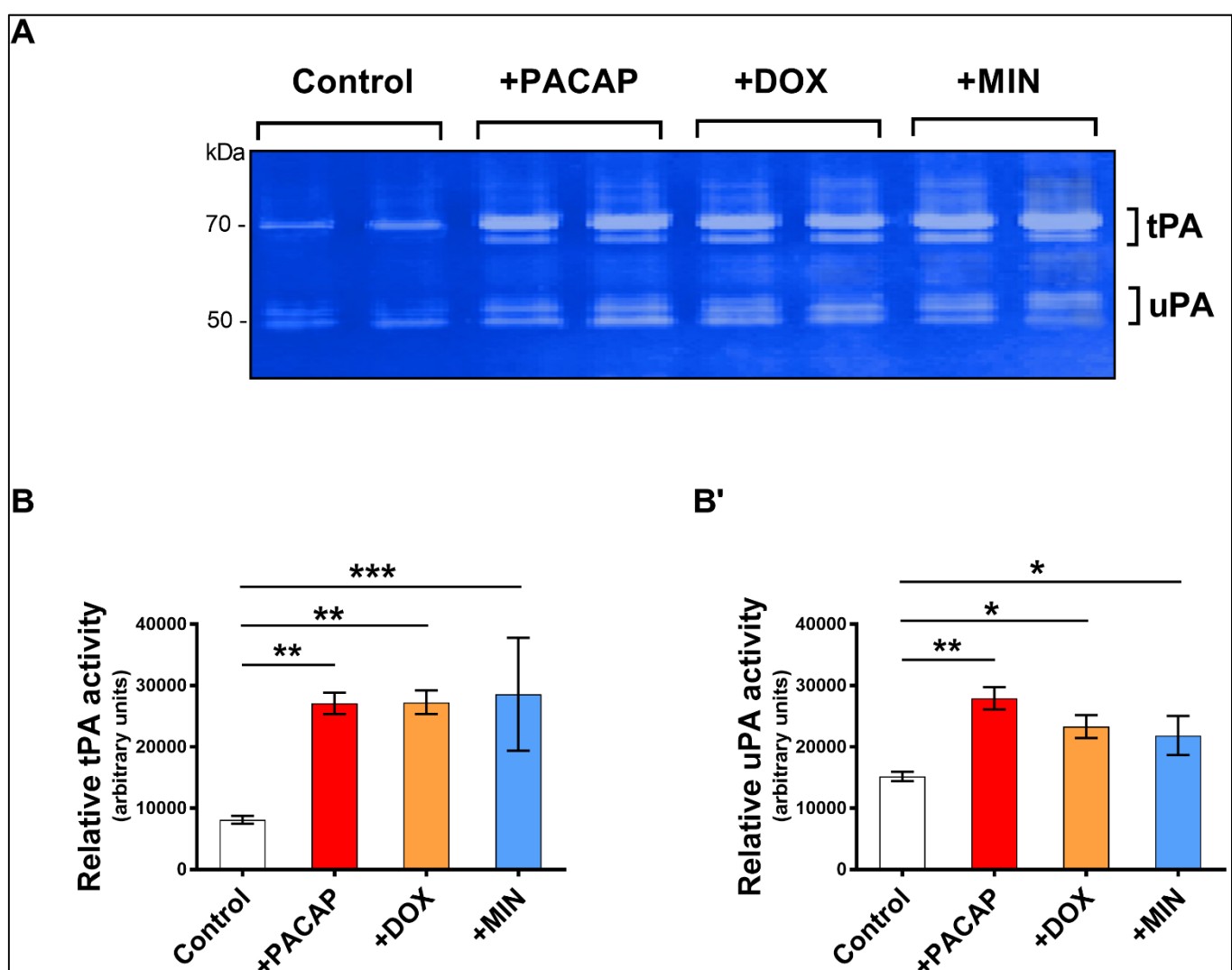

**Figure 3.** Zymographic assessment of tPA and uPA activity of RT4 cells in response to PACAP, doxycycline or minocycline. (**A**) Representative zymogram and densitometric analyses showing the relative levels of (**B**) tPA and (**B′**) after treatment with PACAP, doxycycline or minocycline for 24 h. Values reported (arbitrary units) are the mean $\pm$ S.E.M. of two separate experiments, each run using distinct batches of cells (n = 4/group), measured using ImageJ software. * $p < 0.05$, ** $p < 0.01$ or *** $p < 0.001$, as determined by ANOVA followed by Dunnett's *post-hoc* test.

tPA-like immunoreactivity (IR) was moderate (about 25% of cell surface area) and mainly cytoplasmic in untreated cells (Figure 4B). As expected, administration of PACAP significantly increased the expression of tPA-IR (*** $p < 0.001$ vs. Control, > 40% cell surface area, Figure 4C). Similarly to PACAP, both doxycycline and minocycline increased the percentage of tPA$^+$ IR compared with controls (** $p < 0.01$ vs. control, about 35% of cell surface area, Figure 4D,E).

### 3.5. Pre-Treatment with PACAP6-38 Blocks Doxycycline- and Minocycline-Induced tPA Activity

To assess if doxycycline- and minocycline-induced tPA and uPA activities were related to PAC1 receptor activation, RT4 cells pre-treated (or not) with PACAP6-38 (a PAC1 receptor antagonist, 10 μM) were supplemented with PACAP (100 nM), doxycycline (100 ng/mL) or minocycline (100 ng/mL) for 24 h and enzymatic activities were assessed using zymography.

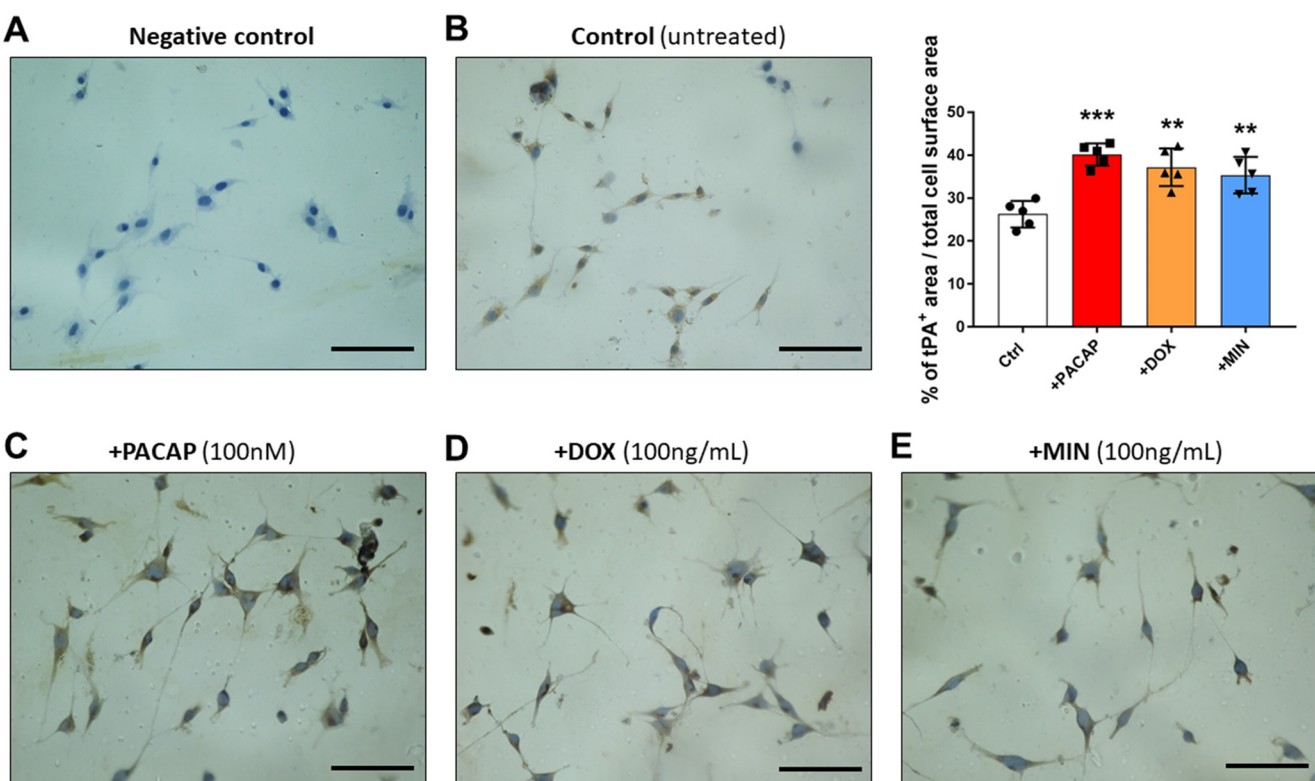

**Figure 4.** Immunocytochemistry of tPA in RT4 cells exposed to PACAP, doxycycline or minocycline. Representative cytochemistry images showing tPA immunoreactivity in (**A**) negative control (no primary Ab), (**B**) untreated, (**C**) PACAP treated, (**D**) doxycycline treated and (**E**) minocycline treated RT4 cells. Values are reported as percentage of tPA-positive cell surface area normalized to total cell surface area $\pm$ S.E.M. ** $p < 0.01$ or *** $p < 0.001$, as determined by ANOVA followed by Dunnett's *post-hoc* test. Magnification = 20×. Scale bar = 60 μm.

As shown, lytic areas on gels showed a significant increase in tPA and uPA activities after treatment with PACAP (** $p < 0.01$ vs. Control for both tPA and uPA), doxycycline (* $p < 0.05$ for both tPA and uPA) and minocycline (** $p < 0.01$ for tPA and * $p < 0.05$ for uPA) (Figure 5A,B,B'). Pre-treatment with PACAP6-38 completely prevented PACAP-induced tPA or uPA activities ($\$\$$ $p < 0.01$ vs. PACAP-treated cells). In cells supplemented with doxycycline or minocycline, PACAP6-38 pre-treatment significantly reduced tPA ($\$$ $p < 0.05$ vs. doxycycline or minocycline treated cells, respectively) and uPA ($\$$ $p < 0.05$ vs. doxycycline-treated and $\$\$$ $p < 0.01$ vs. minocycline treated cells) (Figure 5A,B,B').

*3.6. Pre-Treatment with PACAP6-38 Blocks Doxycycline- and Minocycline-Induced tPA and uPA Protein Expression Levels*

To appraise if pre-treatment with the PAC1 receptor antagonist PACAP6-38 prevented doxycycline and/or minocycline-induced expression of plasminogen activators, we performed Western blot analyses in RT4 cells that were treated as per zymographic analyses.

A statistical significant increase in tPA protein expression was observed both in PACAP- (* $p < 0.05$ vs. Ctrl), doxycycline- or minocycline-treated cells (* $p < 0.05$ for both, Figure 6A,A').

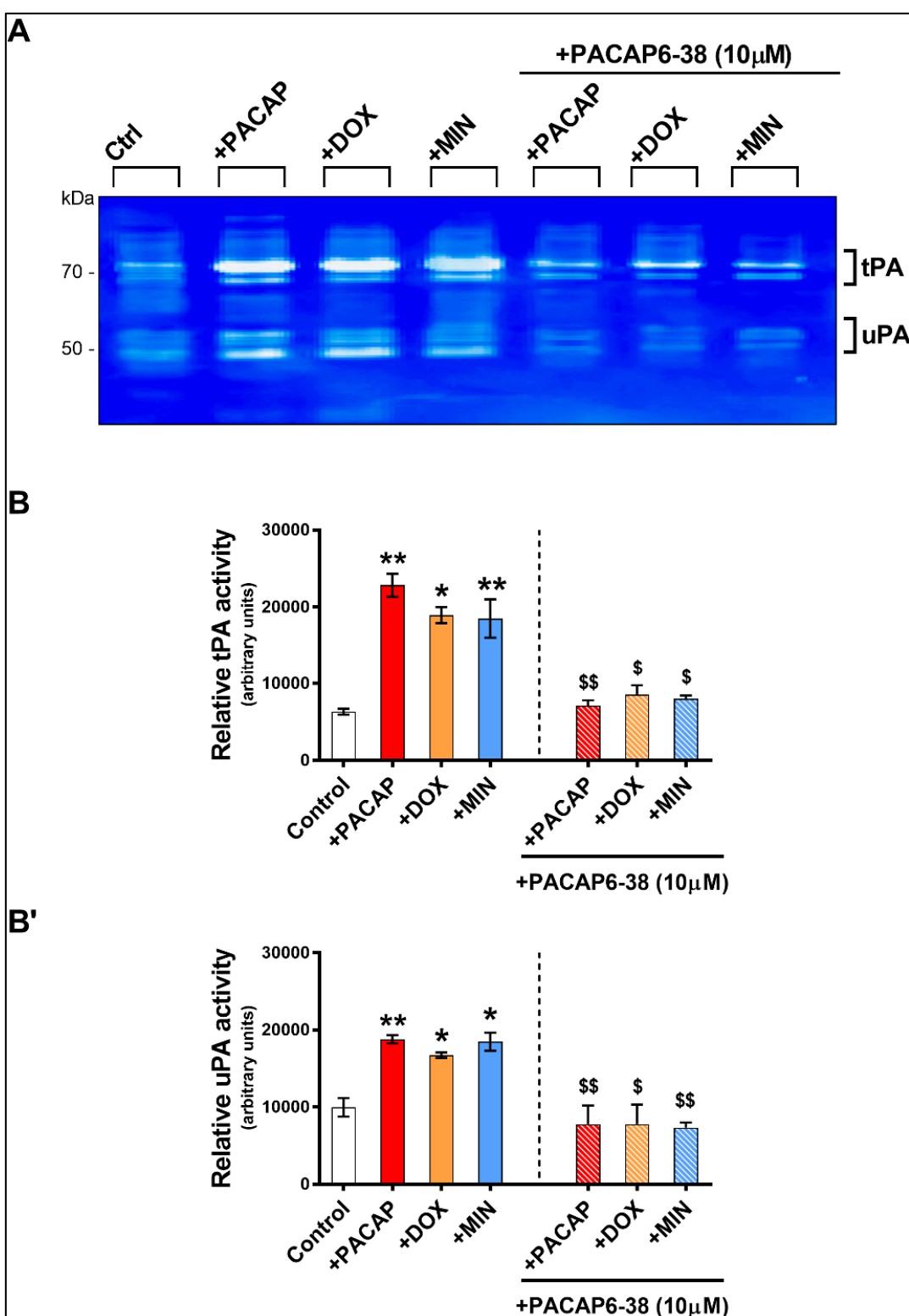

**Figure 5.** Zymographic assessment of tPA and uPA activity of RT4 cells in response to PACAP, doxycycline or minocycline with and without PACAP6-38 pre-treatment. (**A**) Representative zymogram and densitometric analyses showing the relative levels of tPA (**B**) and uPA (**B′**). Cells were treated with either PACAP, doxycycline or minocycline only or pre-treated for 1 h with PACAP6-38 and then assessed after 24 h. Values reported (arbitrary units) are the mean ± S.E.M. of two separate experiments, each run using four separate batches of cells (n = 4 / group), measured using ImageJ software. * $p < 0.05$ or ** $p < 0.01$ vs. Control; \$ $p < 0.05$ or \$\$ $p < 0.01$ vs. corresponding drug-treatment groups, as determined by ANOVA followed by Sidak's *post-hoc* test.

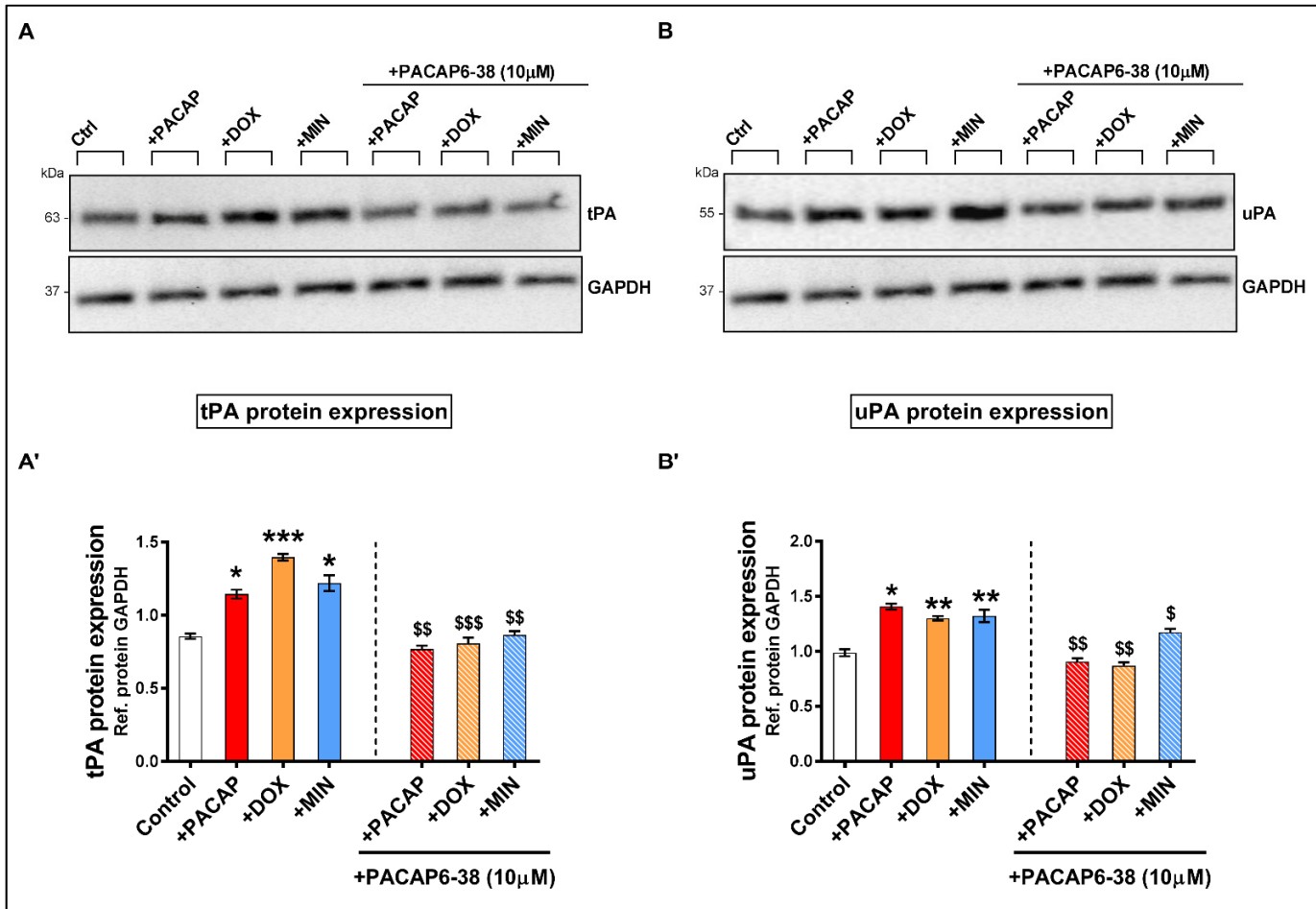

**Figure 6.** Protein expression of tPA and uPA in treated RT4 cells. Western blots and densitometric analyses of tPA (**A**,**A'**) and uPA (**B**,**B'**) in RT4 cells exposed to PACAP6-38 for 1 h then treated with PACAP, doxycycline or minocycline for 24 h. Total proteins were normalized to GAPDH, the loading control. Data represent means of n = 4 samples for each group. Results are expressed as mean ± S.E.M. * $p < 0.05$, ** $p < 0.01$ or *** $p < 0.001$ vs. Control; $ $p < 0.05$, $$ $p < 0.01$ or $$$ $p < 0.001$ vs. corresponding drug-treatment groups, as determined by ANOVA followed by Sidak's *post-hoc* test.

Pre-treatment with PACAP6-38 prevented the induction of tPA expression by PACAP ($$ $p < 0.01$ vs. PACAP-treated), doxycycline ($$$ $p < 0.001$ vs. doxycycline-treated) and minocycline ($$ $p < 0.01$ vs. minocycline-treated). Similar effects were seen when analyzing uPA protein expression. Specifically, PACAP6-38 completely rescued the induction of uPA protein levels caused by PACAP or doxycycline treatment ($$ $p < 0.01$ vs. corresponding drug treatment, Figure 6B,B'). However, PACAP6-38 only partially reduced uPA protein expression in RT4 cells that were treated with minocycline ($ $p < 0.05$ vs. minocycline-treated, Figure 6B,B').

## 4. Discussion

In the present study, we report for the first time that the antibiotics doxycycline and minocycline induce the expression and activity of two plasminogen activators, tPA and uPA, in RT4 Schwann cell-like cultures. Furthermore, we linked this induction to PAC1 receptor activity, as pre-treatment with the PAC1 receptor antagonist, PACAP6-38, mitigated this response. As such, this study provided additional evidence to support the recent finding suggesting that both tetracyclines may act as PAMs of the PAC1 receptor, which was first documented by Song and colleagues in 2019 [23].

The entire study was carried out using RT4 Schwann cells. These immortalized Schwann cell lines share similar transcriptional profiles and biology with primary Schwann cells [32], hence they are considered as a viable alternative to study PNS myelin-producing glia.

Our rationale to conduct this study was based on previous observations, in which we identified that both PACAP and the PAC1 receptors are endogenously expressed in RT4 Schwann cell lines [21]. Using these cells, we have shown that exogenous administration of PACAP upregulates the expression and activity of tPA via the PAC1 receptor and that conditions able to increase the endogenous PACAP/PAC1 signaling (i.e., serum starvation) can also increase the activity of plasminogen activators [22]. In addition, we have also determined that inflammatory factors to mimic the local microenvironment of nerve injury (lipopolysaccharide, aka LPS) can upregulate PACAP expression and concurrently downregulate a direct repressor of plasminogen activators expression (miR-340) in RT4 cells [5], corroborating the idea that the crosstalk between the PACAP neuropeptide system and the plasminogen activator system may be a critical step in the cascade of events that are initiated by Schwann cells following nerve injury. These findings align with other studies showing that these glial cells undergo important biochemical alterations that allow cells to secrete factors able to promote neuronal survival and axonal regeneration after nerve injury [33].

In the RT4 cell line, PACAP upregulates the expression of multiple myelin markers [34], an effect that suggests a broader involvement of the peptide in the nerve regeneration process after injury, summarized here in two distinct stages: (1) an early stage in which PACAP stimulates Schwann cells to secrete factors that promote debris clearance and axonal regrowth and (2) a late stage where PACAP promotes the accumulation of pro-myelinating factors by Schwann cells that aid in remyelination process once neurite regrowth is completed. This theory has been partly confirmed by recent evidence in vivo, where PACAP was found to promote myelin regeneration following sciatic nerve injury [6]. PACAP is significantly increased in peripheral neurons post-injury, and it is detectable at higher levels one month after nerve damage, suggesting that it may play a key role in both axonal regrowth and the consequent remyelination process [35]. Moreover, axon regeneration following injury is severely disrupted in PACAP knockout mice [20]. More recently, transcriptional profiling of the skin of patients undergoing surgery for carpal tunnel syndrome revealed that *ADCYAP1* (the gene encoding for PACAP) was the most robustly upregulated and its expression was associated with nerve recovery. Additionally, when human induced pluripotent stem cell-derived sensory neurons were treated with PACAP, the peptide enhanced axonal outgrowth in a dose dependent manner [36]. Together, these studies suggest a prominent role of the PACAP/PAC1 axis in mediating neuronal survival, axon regeneration and remyelination following peripheral nerve injury.

Targeting the PAC1 receptor has proven to be challenging [15]. Most studies utilize the neuropeptide PACAP to stimulate receptor activation; nonetheless, as with most peptides, PACAP has limited bioavailability and is prone to rapid enzymatic degradation [37]. Despite being able to cross the blood brain barrier, albeit by passive diffusion or through the specific peptide transporter system 6 (PTS6), PACAP has an extremely short half-life [37]. In C57BL/6 mice, it was determined that the half-life of PACAP38 administered intravenously was less than two minutes [38]. Dipeptidyl peptidase IV (DPP IV) is the main enzyme involved in the degradation of PACAP, with mice lacking DPP IV showing significantly slower clearance of the peptide [38,39]. Despite these challenges, PACAP has a safe and high therapeutic index and low doses of PACAP are still able to exert neuroprotection and other beneficial activities.

The recent discovery of doxycycline and minocycline as putative PAMs of the PAC1 receptor may provide an excellent opportunity to target the PAC1 receptor therapeutically. The two antibiotics have well-established neuroprotective and anti-inflammatory functions in the CNS [26,40] and some evidence suggests that minocycline, at least in part, may aid in nerve regeneration in bio-artificial nerve grafts that are free from Wallerian degeneration [41]. However, a few studies have shown that antibiotic treatment can inhibit

Wallerian regeneration [42], an essential process of nerve rejuvenation, whereas others do not [43]. Therefore, it is not clear if the balance between beneficial and detrimental outcomes of tetracycline's treatment can ultimately result in an improved nerve recovery or not. Our data suggest that the upregulation of plasminogen activators in RT4 Schwann cells implicates doxycycline and minocycline as agents that can improve extracellular matrix/debris clearance and consequently, promote axonal regrowth. However, additional work is required to confirm such beneficial activities using primary Schwann cells, neuronal/glial co-cultures and/or in vivo, as the repurposing of this class of antibiotics could indeed become a game changer in the treatment of nerve injury.

## 5. Conclusions

In conclusion, we have confirmed that pharmacological blockade of the PAC1 receptors (and perhaps VPAC receptor subtypes) by PACAP6-38 prevents tetracycline-induced activation of plasminogen activators by RT4 Schwann cells. Whilst the study was not aimed at identifying the existence of a PAC1 binding site for doxycycline or minocycline, the biochemical evidence provided supports their novel function as PAMs of the PAC1-receptor in RT4 Schwann cells. The identification of these tetracyclines as putative PAC1 PAMs may offer a therapeutic advantage for the treatment of peripheral nerve pathologies or injuries where Schwann cells' mediated rejuvenating activities are required.

**Author Contributions:** Conceptualization, A.C.; methodology, S.T.B. and G.M.; formal analysis, S.T.B. and A.C.; investigation, A.C. and S.T.B.; data curation, S.T.B. and A.C.; writing—original draft preparation, S.T.B. and A.C.; writing—review and editing, A.C. and G.M.; supervision, A.C.; project administration, A.C.; funding acquisition, A.C. All authors have read and agreed to the published version of the manuscript.

**Funding:** These experiments were supported by the University of Technology Sydney Start-up Research Fund 2018 and the UTS Seed Fund 2020 to Associate Professor Castorina.

**Institutional Review Board Statement:** Not applicable.

**Conflicts of Interest:** The authors declare no conflict of interest.

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
