# Peer review of "Doxycycline and Minocycline Act as Positive Allosteric Modulators of the PAC1 Receptor and Induce Plasminogen Activators in RT4 Schwann Cells"

_applsci, doi:10.3390/app11167673_

Round 1

Reviewer 1 Report

In this manuscript, the authors have demonstrated that doxycline or minocycline treatment induces tPA and uPA via activating PAC1 by various means, i.e. QPCR, zymography, western blotting. The experiments were well-designed, performed and properly analyzed. Only some minor points need to be addressed.

  1. In figure 1, the authors showed that doxycline (100ng/ml) and minocycline (100ng/ml) induce the highest expression of tPA and uPA. Please explain why the expressions of tPA and uPA are lower upon stimulation of higher doxycline and minocycline.
  2. In figure 2, please include the statistical analysis among different groups to demonstrate whether cell viability is affected by DOX and MIN at high concentration.
  3. In figure 5, the authors used PACAP6-38 as a PAC1 antagonist. However, the concentration of PACAP6-38 employed in this experiment was much higher than its IC50, causing the possible problem of unspecificity. The author should address this question by titrating PACAP6-38 or introducing PAC1 knockdown. 

Reviewer 2 Report

This paper by Broome et al is a straight forward attempt to explore one of the reasons behind neuroprotection and axonal regeneration. The authors have presented data sets from conventional bio-analytical experiments to show a plausible over-expression of tPA and uPA which they attribute to antibiotic effect. The data set is nice. But there are a few concerns especially the data presentation, english language and grammar, low resolution immunocytochemistry images, lack of sufficient novelty. The authors need to explain more especially how this data could lead to development of novel therapeutic solutions to address diverse neurological disorders. 
